# *IFI16* Induced by Direct Interaction between Esophageal Squamous Cell Carcinomas and Macrophages Promotes Tumor Progression via Secretion of IL-1α

**DOI:** 10.3390/cells12222603

**Published:** 2023-11-10

**Authors:** Yuki Azumi, Yu-ichiro Koma, Shuichi Tsukamoto, Yu Kitamura, Nobuaki Ishihara, Keitaro Yamanaka, Takashi Nakanishi, Shoji Miyako, Satoshi Urakami, Kohei Tanigawa, Takayuki Kodama, Mari Nishio, Manabu Shigeoka, Yoshihiro Kakeji, Hiroshi Yokozaki

**Affiliations:** 1Division of Pathology, Department of Pathology, Kobe University Graduate School of Medicine, Kobe 650-0017, Japan; 202m856m@gsuite.kobe-u.ac.jp (Y.A.); stsuka@med.kobe-u.ac.jp (S.T.); ishihara@med.kobe-u.ac.jp (N.I.); 213m889m@gsuite.kobe-u.ac.jp (K.Y.); 215m857m@stu.kobe-u.ac.jp (T.N.); shoji224@med.kobe-u.ac.jp (S.M.); urasato@med.kobe-u.ac.jp (S.U.); takodama@med.kobe-u.ac.jp (T.K.); marin@med.kobe-u.ac.jp (M.N.); mshige@med.kobe-u.ac.jp (M.S.); hyoko@med.kobe-u.ac.jp (H.Y.); 2Division of Gastro-Intestinal Surgery, Department of Surgery, Kobe University Graduate School of Medicine, Kobe 650-0017, Japan; y.kitamura-0916@people.kobe-u.ac.jp (Y.K.); 188m863m@gsuite.kobe-u.ac.jp (K.T.); kakeji@med.kobe-u.ac.jp (Y.K.); 3Division of Hepato-Biliary-Pancreatic Surgery, Department of Surgery, Kobe University Graduate School of Medicine, Kobe 650-0017, Japan; 4Division of Obstetrics and Gynecology, Department of Surgery Related, Kobe University Graduate School of Medicine, Kobe 650-0017, Japan; 5Division of Gastroenterology, Department of Internal Medicine, Kobe University Graduate School of Medicine, Kobe 650-0017, Japan

**Keywords:** esophageal squamous cell carcinoma, tumor-associated macrophage, direct co-culture, *IFI16*, IL-1α

## Abstract

Tumor-associated macrophages (TAMs), one of the major components of the tumor microenvironment, contribute to the progression of esophageal squamous cell carcinoma (ESCC). We previously established a direct co-culture system of human ESCC cells and macrophages and reported the promotion of malignant phenotypes, such as survival, growth, and migration, in ESCC cells. These findings suggested that direct interactions between cancer cells and macrophages contribute to the malignancy of ESCC, but its underlying mechanisms remain unclear. In this study, we compared the expression levels of the interferon-induced genes between mono- and co-cultured ESCC cells using a cDNA microarray and found that interferon-inducible protein 16 (*IFI16*) was most significantly upregulated in co-cultured ESCC cells. *IFI16* knockdown suppressed malignant phenotypes and also decreased the secretion of interleukin-1α (IL-1α) from ESCC cells. Additionally, recombinant IL-1α enhanced malignant phenotypes of ESCC cells through the Erk and NF-κB signaling. Immunohistochemistry revealed that high *IFI16* expression in human ESCC tissues tended to be associated with disease-free survival and was significantly associated with tumor depth, lymph node metastasis, and macrophage infiltration. The results of this study reveal that *IFI16* is involved in ESCC progression via IL-1α and imply the potential of *IFI16* as a novel prognostic factor for ESCC.

## 1. Introduction

Esophageal cancer is a highly malignant neoplasm and the sixth most common cause of cancer-related deaths worldwide [1], with the highest incidence in East Asia [2]. Among the two most common histological subtypes, esophageal squamous cell carcinoma (ESCC) and adenocarcinoma, ESCC accounts for approximately 90% of all esophageal cancer cases in Japan [3,4]. Despite multidisciplinary treatments, including surgical resection, chemotherapy, and radiotherapy, the prognosis of ESCC remains poor, warranting an elucidation of the underlying mechanisms in the pathophysiology of highly malignant forms [3,5].

Immune checkpoint inhibitors (ICIs) have been shown to improve prognosis in several cancers, including ESCC [6,7,8,9]. Their mechanism of action relies on the activation of the antitumor function of T cells [10,11]. However, the interaction of cancer cells and stromal cells has been reported to influence clinical outcomes following ICI therapy [10,11,12,13]. This highlights the key role of the tumor microenvironment (TME), which is the site of interaction of cancer and stromal cells in cancer treatment. The TME is composed of various stromal cells, including leukocytes such as lymphocytes and macrophages, as well as fibroblasts [14]. Macrophages present in the TME are called tumor-associated macrophages (TAMs) [15]. We previously reported the association of CD204-positive TAMs with poor prognosis and cancer progression in patients with ESCC [16]. Many humoral factors that interact with ESCC cells and TAMs have been reported using an indirect co-culture system [17,18,19]. In the actual disease state, there is a direct contact between TAMs and cancer cells. Therefore, we established a direct co-culture system between human ESCC cells and peripheral blood-derived macrophages to simulate the TME. We investigated the changes in gene expression levels between mono-culture ESCC cells and ESCC cells directly co-cultured with macrophages [20]. Among the genes upregulated in co-cultured ESCC cells, we reported that S100 calcium-binding proteins A8 and A9 (S100A8/A9), interleukin-7 receptor (IL-7R), and matrix metalloproteinase 9 (MMP9) are involved in tumor progression and poor prognosis [20,21,22]. The S100A8/A9 complex of the pro-inflammatory cytokine S100 family enhanced the migration and invasion of ESCC cells by activating the Akt and p38 MAPK pathways. IL-7R, one of the interleukin-related molecules, promoted the survival and growth of ESCC cells via the Akt and Erk pathways. MMP9, a zinc-dependent protease, was also reported to facilitate the migration and invasion of ESCC cells.

Interferons are also the most well-known cytokine family. In addition to their functions in the immune response, they are involved in cancer progression [23]. Several pathways driven by interferons have been reported to regulate the expression of genes encoding proteins involved in tumor progression and immune cell regulation [23,24]. Therefore, an analysis of the factors mediating the interaction between interferons and cancers is necessary to further understand tumor progression. The human HIN-200 (hematopoietic, interferon-inducible nuclear proteins with a 200 amino acid repeat) family is a group of interferon-induced genes, with each protein possessing either one or two 200 amino-acid sequence domains at the C-terminus that mediate protein–protein interactions [25]. This group includes interferon-inducible protein 16 (*IFI16*), myeloid cell nuclear differentiation antigen (MNDA), absent in melanoma 2 (AIM2), and pyrin and HIN domain family member 1 (PYHIN1), each of which has been reported to be related to tumor progression [26,27,28,29]. In this study, we focused on the role of the HIN200 family in the malignant phenotype enhancement of ESCC cells through direct co-culture with macrophages.

## 2. Materials and Methods

### 2.1. Cell Lines and Cell Culture

The human ESCC cell lines TE-9, TE-10, and TE-11 (poorly, highly, and moderately differentiated type, respectively) were purchased from the cell bank of RIKEN Bioresource Center (Tsukuba, Japan). The cell lines were cultured in an RPMI-1640 medium (FUJIFILM Wako Chemicals, Osaka, Japan) supplemented with 10% fetal bovine serum (FBS; Sigma-Aldrich, St. Louis, MO, USA) and a 1% antibiotic–antimycotic mixed stock solution (FUJIFILM Wako Chemicals) at 37 °C with 5% CO_2_.

### 2.2. Establishment of Human Peripheral Blood-Derived Macrophages

Peripheral blood-derived macrophages were established as previously described [20]. Briefly, peripheral blood samples were collected from the healthy volunteers. Peripheral blood layered on Ficoll-Paque^TM^ PREMIUM (Cytiva, Chicago, IL, USA) was centrifuged, and the resulting buffy coat was collected and mixed with anti-CD14 microbeads (130-050-201; Miltenyi Biotec, Bergisch Gladbach, Germany). CD14-positive peripheral blood monocytes (PBMos) were isolated using an autoMACS^®^ Pro Separator (Miltenyi Biotec). Next, 1 × 10^6^ PBMos were cultured in RPMI-1640 with 10% FBS, 1% antibiotic-antimycotic, and 25 ng/mL recombinant human M-CSF protein (rhM-CSF; R&D Systems, Minneapolis, MN, USA) in a 10 cm dish. After a six-day incubation period, the cells differentiated into macrophages.

### 2.3. Direct Co-Culture System between ESCC Cells and Macrophages

Direct co-culture was performed as previously described [20]. Subsequently, cultured macrophages were washed three times with FBS-free RPMI-1640. Then, 2 × 10^6^ ESCC cells (TE-9, TE-10, and TE-11) suspended in FBS-free RPMI-1640 were seeded onto the macrophages or seeded in a macrophage-free 10 cm dish and incubated for 48 h to establish co-cultured or mono-cultured ESCC cells, respectively. The cells were then washed thrice with phosphate-buffered saline (PBS; FUJIFILM Wako Chemicals) and detached from the dish using trypsin (FUJIFILM Wako Chemicals). The collected cells were mixed with anti-EpCAM microbeads (130-061-101; Miltenyi Biotec), and tumor cells with a high purity were separated using an autoMACS^®^ Pro Separator.

### 2.4. cDNA Microarray

Previously, a cDNA microarray was conducted on mono-culture TE-11 cells and TE-11 cells that were directly co-cultured with macrophages [20]. The data deposited in the Gene Expression Omnibus database (GSE174796) were reexamined.

### 2.5. Quantitative Real-Time Polymerase Chain Reaction (qRT-PCR) and Reverse Transcription Polymerase Chain Reaction (RT-PCR)

Total RNA was isolated from cells using an RNeasy kit (Qiagen, Hilden, Germany). The cDNA was synthesized for qRT-PCR from RNA using a ReverTra Ace^®^ qPCR RT master Mix (TOYOBO, Osaka, Japan). qRT-PCR was performed using the StepOnePlus Real-Time PCR System (Applied Biosystems, Foster City, CA, USA) and Fast SYBR^TM^ Green Master Mix (Applied Biosystems) along with specific primers of each target gene. The mRNA expression levels of the targets in the samples were quantified using the comparative threshold cycle (C_T_) method according to the manufacturer’s instructions, as in our previous studies [16,19,20]. RT-PCR was performed using the QIAGEN OneStep RT-PCR kit (Qiagen) for 40 cycles at an annealing temperature of 60 °C and was then separated by electrophoresis using a 2% agarose gel. *GAPDH* was quantified as an internal control. The primer sequences of the targets for qRT-PCR and RT-PCR were as follows: *IFI16*, 5′-TAGAAGTGCCAGCGTAACTCC-3′ (forward), 5′-TGATTGTGGTCAGTCGTCCA-3′ (reverse); *IL1A*, 5′-AGATGCCTGAGATACCCAAACC-3′ (forward), 5′-CCAAGCACACCAGTAGTCT-3′ (reverse); *IL1R1*, 5′-TGCCTGAGGTCTTGGAAAAAC-3′ (forward), 5′-TGTGGTCCCTGTGTAAAGTCC-3′ (reverse); *GAPDH* (for qRT-PCR), 5′-GCACCGTCAAGGCTGAGAAC-3′ (forward), 5′-ATGGTGGTGAAGACGCCAGT-3′ (reverse); *GAPDH* (for RT-PCR), 5′-ACCACAGTCCATGCCATCAC-3′ (forward), and 5′-TCCACCACCCTGTTGCTGTA-3′ (reverse).

### 2.6. Western Blotting

The cultured cells were washed with cold PBS (4 °C) and lysed with lysis buffer (50 mmol/L Tri-HCl at pH 7.5 with 125 mmol/L NaCl, 5 mmol/L EDTA, and 0.1% Triton X-100) containing 1% protease inhibitor cocktail and 1% phosphatase inhibitor cocktail (Sigma-Aldrich), as we previously performed [20]. The lysed cells were agitated for 30 min and then centrifuged at 10,000× *g* for 10 min at 4 °C. Concentrations of the extracted proteins were measured using NanoDrop Lite (Thermo Fisher Scientific, Waltham, MA, USA). Each protein was loaded onto a 5–20% gradient sodium dodecyl sulfate–polyacrylamide gel for electrophoresis and then transferred to a polyvinylidene difluoride membrane using an iBlot2 system (Invitrogen, Carlsbad, CA, USA). The membrane was blocked with 5% skim milk and incubated with the primary antibody of the target protein at 4 °C for 24–48 h. After incubation with the appropriate secondary antibody for 90 min at room temperature, the membrane was incubated with ImmunoStar reagent (FUJIFILM Wako Chemicals). The bands were visualized using an ImageQuant^TM^ LAS4000 mini (FUJIFILM, Tokyo, Japan).

The primary antibodies used were *IFI16* (Cell Signaling Technology; CST, Beverly, MA, USA), phospho-(p) NF-κB p65 (#3033, CST), NF-κB p65 (#8242, CST), pErk1/2 (#9101, CST), Erk1/2 (#4695, CST), IL-R1 (sc-393998, Santa Cruz Biotechnology, Dallas, TX, USA), and β-actin (#4970, CST). The secondary antibodies were horseradish peroxidase (HRP)-conjugated anti-rabbit IgG (#NA934V; Cytiva) and HRP-conjugated anti-mouse IgG (#NA931V; Cytiva).

### 2.7. Knockdown of IFI16 in ESCC Cells

ESCC cells were transfected with small interfering RNA (siRNA) targeting human *IFI16* (si*IFI16*; 20 nM; sc-35633; Santa Cruz Biotechnology) using Lipofectamine RNAiMAX (Invitrogen). MISSION^®^ siRNA Universal Negative Control #1 (siNC; 20 nM; Sigma-Aldrich) was used as a negative control. The cells were cultured for 48 h before use in experiments.

### 2.8. MTS Assay

To determine their survival and growth, 1 × 10^4^ ESCC cells per well were seeded in FBS-free RPMI-1640 in 96-well plates, and 5 × 10^3^ ESCC cells per well were seeded in RPMI-1640 supplemented with 1% FBS, respectively. After 24 or 48 h, 20 μL of CellTiter 96 Aqueous One Solution Reagent (Promega, Madison, WI, USA) was added to each well. The plates were incubated for 60 min at 37 °C. The absorbance was measured at 492 nm using an Infinite 200 PRO microplate reader (Tecan, Mannedorf, Switzerland). In some experiments, ESCC cells were treated with 1 μg/mL neutralizing antibody against IL-1α (AF-200-NA, R&D Systems) or 1 μg/mL normal goat IgG (AB-108-C, R&D Systems) as the negative control, with or without 10 ng/mL rhIL-1α; 10 ng/mL rhIL-1α with Erk inhibitor (PD98059, CST), NF-κB inhibitor (Bay117082; Sigma-Aldrich), or Dimethyl Sulfoxide (DMSO; FUJIFILM Wako Chemicals) as the negative control.

### 2.9. Transwell Migration Assay

Cell culture insert (8.0 μm pore size, BD Falcon, Lincoln Park, NY, USA) was seeded with 1 × 10^5^ ESCC cells per well in 300 μL of FBS-free RPMI-1640 (the upper chamber). The chambers were then placed in each well of a 24-well plate containing 800 μL of RPMI-1640 supplemented with 0.1% FBS (the lower chamber). After incubation for 24 or 48 h at 37 °C, the number of cells that migrated to the lower membrane was counted. In certain experiments, ESCC cells were treated with 1 μg/mL neutralizing antibody against IL-1α (AF-200-NA, R&D Systems), or 1 μg/mL normal goat IgG (AB-108-C, R&D Systems) as the negative control was applied to each well, with or without 10 ng/mL rhIL-1α; 10 ng/mL rhIL-1α with Erk inhibitor (PD98059, CST), NF-κB inhibitor (Bay117082; Sigma-Aldrich), or DMSO (FUJIFILM Wako Chemicals) as the negative control.

### 2.10. Cytokine Array

A total of 1.5 × 10^6^ ESCC cells transfected with siRNA were seeded into a 6-well plate with 3 mL of FBS-free RPMI-1640 media for 24 h. The culture supernatants were collected and analyzed. The Proteome Profiler Human XL Cytokine Array Kit (R&D Systems) was used to compare the culture supernatant of TE-11 cells transfected with si*IFI16* or siNC, following the manufacturer’s instructions.

### 2.11. Enzyme-Linked Immunosorbent Assay (ELISA)

The culture supernatants of ESCC cells were prepared as described above. The concentration of IL-1α in TE-9, TE-10, and TE-11 cells was measured using the Human IL-1 alpha/IL-1F1 Quantikine ELISA Kit (R&D Systems) following the manufacturer’s instructions.

### 2.12. ESCC Tissue Samples

Surgically resected ESCC tissue samples were collected at the Kobe University Hospital, Japan. After excluding patients who received preoperative therapy (chemotherapy and/or radiation), 69 patients were included in the analysis. Clinical data and pathological diagnoses were analyzed based on the 10th edition of the Japanese Classification of Esophageal Cancer [30,31] and the 7th edition of the Union for International Cancer Control (UICC) TNM Classification of Malignant Tumours [32]. All patients provided informed consent for the use of their resected samples for research purposes. This study was conducted in accordance with the Declaration of Helsinki and approved by the Kobe University Institutional Review Board (B210103).

### 2.13. Immunohistochemistry

Immunohistochemical analyses of 4 μm-thick tissue sections were performed on a BOND-MAX automated system (Leica Biosystems, Bannockburn, IL, USA) using a BOND Polymer Refine Detection Kit (Leica Biosystems). The antibody against *IFI16* (CST) was used at a dilution of 1:400. The strongest staining intensity in the invasive area of the tumor was classified as high or low. An equal or stronger intensity of ESCC cells compared with the basal cells of the adjacent non-tumoral tissue was classified as *IFI16*-high, and a weaker intensity compared with the basal cells of the adjacent non-tumoral tissue was classified as *IFI16*-low. Based on the aforementioned criteria, all tissue samples were evaluated by two expert pathologists (Y.-i.K. and H.Y.) and one surgeon (Y.A.).

### 2.14. Statistical Analysis

Each in vitro experiment was performed in triplicate. Experimental data were analyzed using a two-tailed Student’s *t*-test. Clinicopathological data were analyzed using Chi-Squared (χ^2^) tests. Survival curves for overall, cause-specific, and disease-free survival were constructed using the Kaplan–Meier survival analysis. The results of the two groups were compared using the log-rank test. Statistical significance was set at *p* < 0.05. The statistical analysis was performed using the software IBM SPSS Statistics 22 (IBM Corp., Armonk, NY, USA).

## 3. Results

### 3.1. Direct Co-Culture with Macrophages Induces Upregulation of IFI16 Expression and Promotion of Erk and NF-κB Signaling Pathways in ESCC Cells

To elucidate the role of the HIN-200 family in ESCC, we reviewed our previous cDNA microarray results (GSE174796) comparing mono-cultured ESCC cells and co-cultured ESCC cells with macrophages. Within the HIN-200 gene family, *IFI16* emerged as the most significantly upregulated gene in the co-cultured ESCC cells compared to the mono-cultured ESCC cells (Table 1). Consequently, we chose to investigate the impact of *IFI16* on the malignancy of ESCC. To confirm the expression of *IFI16* mRNA and protein in the ESCC cell lines (TE-9, TE-10, and TE-11), qRT-PCR and Western blotting were performed, respectively. A significant upregulation of *IFI16* was observed in the directly co-cultured ESCC cell lines (TE-9 co, TE-10 co, and TE-11 co), as shown in Figure 1A,B and Appendix A. We also evaluated the signaling pathways activated via direct co-culture with macrophages and found increased phosphorylation of NF-κB in all the directly co-cultured ESCC cell lines (Figure 1B and Appendix A). We have also previously demonstrated increased Erk phosphorylation in co-cultured ESCC cell lines [21].

### 3.2. IFI16 Knockdown Suppresses the Survival, Growth, and Migration of ESCC Cells through Erk and NF-κB Signaling Pathways

To evaluate the contribution of *IFI16* to enhanced malignant phenotypes in directly co-cultured ESCC cell lines, *IFI16* expression in ESCC cells was silenced using siRNA. The silencing of *IFI16* was confirmed with qRT-PCR and Western blotting (Figure 2A,B and Appendix A). *IFI16* silencing suppressed the phosphorylation of both Erk and NF-κB in the three ESCC cell lines (TE-9, TE-10, and TE-11) compared with the siNC-transfected ESCC cell lines (Figure 2B and Appendix A). In addition, MTS and Transwell migration assays were conducted to evaluate the effect of *IFI16* silencing in the malignant phenotypes of ESCC cell lines. The results demonstrated the suppression of survival, growth, and migration in the *IFI16*-silenced ESCC cell lines compared to the siNC-transfected ESCC cell lines (Figure 2C–E and Appendix A).

### 3.3. IFI16-Regulated IL-1α Secretion from ESCC Cells Plays a Critical Role in the Induction of Malignant Phenotypes Following Direct Co-Culture with Macrophages

*IFI16* is known to be a positive regulator of cytokine secretion [26,33,34]. Therefore, we conducted a cytokine array to identify humoral factors regulated by *IFI16* in TE-11 cells transfected with si*IFI16* or siNC (Appendix A). IL-1α spots were shown to be suppressed by *IFI16*-silencing in the TE-11 cells (Figure 3A). The expression of *IL1A* mRNA was significantly suppressed in the *IFI16*-silenced ESCC cell lines (TE-9, TE-10, and TE-11) compared to that in siNC-transfected ESCC cell lines (Figure 3B). Also, the IL-1α secretion from *IFI16*-silenced ESCC cell lines was significantly suppressed compared to that of siNC-transfected ESCC cell lines (Figure 3C). Moreover, MTS and transwell migration assays using neutralizing antibodies against IL-1α were conducted to assess the role of IL-1α in enhancing the malignant phenotypes of ESCC cells after direct co-culture with macrophages. While migration was significantly suppressed in the three ESCC cell lines by the neutralizing antibody, survival was suppressed only in the TE-9 cells, and growth was suppressed in only the TE-9 and TE-11 cells (Figure 3D–F and Appendix A). The malignant phenotypes of ESCC cell lines promoted by direct co-culture with macrophages were shown to be partially mediated by *IFI16*-regulated IL-1α secretion.

### 3.4. IL-1α Promotes Malignant Phenotypes of ESCC Cells through Erk and NF-κB Signaling Pathways

Since we found that *IFI16* promoted the malignant phenotype of ESCC cells via the secretion of IL-1α, we then investigated the effect of exogenous IL-1α on ESCC cells using rhIL-1α. First, the expression of the IL-1α receptor, IL-1R1, in the ESCC cell lines (TE-9, TE-10, and TE-11) was determined by with qRT-PCR and Western blotting (Appendix A). We further examined the effect of IL-1α through IL-1R1. Next, treatment with rhIL-1α promoted survival and migration in the ESCC cell lines (TE-9, TE-10, and TE-11), but did not affect their growth (Figure 4A–C and Appendix A). The effect of rhIL-1α on the Erk and NF-κB signaling pathways was then evaluated. The RhIL-1α treatment promoted the phosphorylation of the Erk and NF-κB signaling pathways in all the ESCC cell lines (Figure 4D and Appendix A). Furthermore, MTS and transwell migration assays were conducted in the rhIL-1α-treated ESCC cells with an Erk inhibitor (PD98059) or NF-κB inhibitor (BAY11-7082) to evaluate the effect of inhibiting Erk or NF-κB in rhIL-1α-induced malignant phenotypes. Migration in the three ESCC cell lines was abrogated by both inhibitors, whereas survival was abrogated only in the TE-9 and TE-10 cells, and growth was abrogated only in the TE-10 and TE-11 cells (Figure 4E–G and Appendix A).

### 3.5. High Expression of IFI16 in ESCC Tissues Is Associated with Macrophage Infiltration and Poor Prognosis

To verify the correlation between the expression of *IFI16* and the clinical outcomes of patients with ESCC, 69 surgically resected human ESCC tissues were analyzed using immunohistochemistry. The tissues were divided into *IFI16*-high and *IFI16*-low groups depending on the staining intensity of *IFI16* in the invasive area of the tumor (Figure 5A). Forty-five cases were classified into the *IFI16*-high group, and 24 patients were classified into the *IFI16*-low group. The clinicopathological data of the patients in the two groups were compared, and the progressive depth of tumor invasion (*p* = 0.042) and positive lymph node metastasis (*p* = 0.039) significantly correlated with a high expression of *IFI16* (Table 2). In addition, a high expression of *IFI16* was associated with high proportions of CD68- (*p* = 0.009), CD163- (*p* = 0.053), and CD204- (*p* = 0.015) positive macrophages (Table 2). Finally, the long-term prognosis of patients with ESCC in the *IFI16*-high and *IFI16*-low groups was compared using Kaplan–Meier survival analysis. Based on the survival curves, only the disease-free survival rate of the *IFI16*-high group tended to show a worse prognosis compared to the *IFI16*-low group (*p* = 0.083, Figure 5B).

## 4. Discussion

In our previous work, we reported that direct co-culture with macrophages intensifies the malignant properties of ESCC cells, proposing that the underlying mechanisms could reveal new insights into ESCC progression [20,21,22]. In the current study, we have shown that ESCC cells, when directly co-cultured with macrophages derived from peripheral blood, exhibit an increased expression of *IFI16*. This elevation in *IFI16* was associated with a subsequent increase in IL-1α secretion from the ESCC cells. Further, we discovered that IL-1α enhances the survival, growth, and migration of ESCC cells via the NF-κB and Erk signaling pathways, operating in an autocrine fashion. Clinically, high *IFI16* expression levels in patients of ESCC tend to correlate with poorer outcomes in terms of disease-free survival, positioning *IFI16* as a promising prognostic marker for ESCC. The in vitro experimental outcomes are summarized in Figure 6.

*IFI16* is a protein located in the nucleus and cytoplasm and acts as a DNA sensor during viral infections [35,36]. *IFI16* has been reported to bind to DNA viruses such as cytomegalovirus and Kaposi sarcoma-associated herpesvirus, leading to the release of interferon β [37,38]. Recent studies have reported anti-*IFI16* antibodies in the sera of patients with autoimmune diseases such as systemic lupus erythematosus, inflammatory bowel diseases, and rheumatoid arthritis [39,40,41]. Other studies have provided evidence of the involvement of *IFI16* in the cell death pathway in response to cell damage caused by radiation exposure and aging [42]. Several experiments have shown *IFI16* expression in B lymphocytes and macrophages as well as in normal endothelial and epithelial cells [43,44,45]. In macrophages, *IFI16* is essential for the activation of the cGAS-cGAMP-STING-TBK1 pathway upon sensing viral DNA, leading to the induction of interferon secretion [43]. *IFI16* expression in B lymphocytes is inversely correlated with several master regulators of B cell differentiation [44]. However, the role of *IFI16* in normal endothelial and epithelial cells remains unclear [45].

The relationship between *IFI16* and cancer progression has been elucidated via several mechanisms. Experiments have shown that the two HIN domains of *IFI16* enhance p53-mediated p21 activation and inhibit tumor growth by binding to the C-terminus and core domain of p53 [33,46]. In addition, *IFI16* shows antitumor effects by inhibiting DNA repair in tumor cells via STING-induced type I interferon signaling. In contrast, the *IFI16*-induced STING pathway reportedly promotes tumor growth by causing an infiltration of immunosuppressive cells such as regulatory T cells [47,48]. *IFI16* has been reported to form an inflammasome by binding to an NOD-like receptor involved in tumor progression [49,50,51]. The tumor-promoting and tumor-suppressive roles of *IFI16* have been evaluated in various cell lines. The suppressive role of *IFI16* against malignant phenotypes in tumor cells has been reported in hepatocellular carcinoma (HCC), prostate cancer, and thyroid cancer [33,52,53]. In contrast, tumor-promoting roles of *IFI16* have been reported in cervical cancer, breast cancer, and renal cell carcinoma (RCC) [54,55,56]. In this study, our cDNA microarray analysis revealed that *IFI16* expression is upregulated in ESCC cell lines when co-cultured with macrophages. This finding aligns with research by Wang et al., who identified high *IFI16* expression in highly metastatic ESCC cells using a proteomics approach [26]. They characterized *IFI16* as a protein associated with metastasis in ESCC and observed that silencing *IFI16* led to decreased levels of fibroblast growth factor proteins FGF1 and FGF2 in 30M cells, a derivative of the KYSE30 cell line and a model for ESCC metastasis. Moreover, they demonstrated that FGF1 and FGF2 could reverse the suppressive effects of *IFI16* knockdown on the migration and invasion of ESCC cells.

Following the report of FGF proteins [26], since cytokines whose expression were regulated by *IFI16* may be involved in the malignant phenotype of ESCC, we also searched comprehensively for cytokines whose expression is downregulated by *IFI16* knockdown using a cytokine array. This cytokine array also included FGF2 (alternative name FGF basic, C9 and C10 spots in Appendix A), but we could not detect any FGF2 spots in our experiments. This difference may be due to the use of different ESCC cell lines (TE-11). Instead, our study identified IL-1α as another cytokine under the regulatory influence of *IFI16*, in addition to the FGF proteins. The secretion of IL-1α induced by the interaction of *IFI16* with the ASC (apoptosis-associated speck-like protein) following ultraviolet exposure was reported in human keratinocytes [57]. The IL-1 family of cytokines includes IL-1α, IL-1β, IL-1Ra, IL-18, IL-33, IL-36Ra, IL-36α, IL-36β, IL-36γ, IL-37, and IL-38, and they play an important role in the regulation of inflammation [58]. IL-1α binds to IL-1R1, inducing the secretion of proinflammatory cytokines through the activation of the NF-κB and Erk signaling pathways [59]. The involvement of IL-1α in autoimmune diseases and infectious diseases has been reported [60]. Both the tumor-suppressive and tumor-promoting functions of IL-1α on tumors have been demonstrated. Previous studies have shown that IL-1α suppresses tumorigenesis in fibrosarcoma and breast cancer [61,62]. In head and neck cancer and colorectal cancer, IL-1α expression was associated with the promotion of malignant phenotypes. Lin et al. reported that IL-1α exerts immunosuppressive and tumor-promoting effects in HCC [63]. Chen et al. reported that IL-1Ra inhibits ESCC growth by blocking IL-1α [64]. In this study, we demonstrated that *IFI16* induced by the direct interaction of ESCC cells with macrophages regulated the secretion of IL-1α in ESCC cells. IL-1α promoted malignant phenotypes in ESCC cells in an autocrine and paracrine manner. We also confirmed the involvement of the Erk and NF-κB signaling pathways in this process, similar to inflammatory responses [59]. Synthesizing our findings with those of other studies, it seems likely that various molecules might contribute to the malignancy of cancer through the increased expression of *IFI16*. Notably, our research is the first to identify IL-1α as a factor that promotes the progression of ESCC cells under the regulation of *IFI16*.

In this study, typical prognostic factors, tumor invasion and lymph node metastasis, were positively correlated with a high expression of *IFI16* in resected ESCC samples. In addition, high expression of a pan-macrophage surface marker (CD68) and TAM surface markers (CD163 and CD204) was positively correlated with high *IFI16* expression. These results support our in vitro findings that *IFI16* expression is induced by the interaction between ESCC cells and macrophages. In our cohort, patients with ESCC with high *IFI16* expression showed a tendency towards poor disease-free survival. A high expression of *IFI16* in tumors has been significantly associated with poor overall survival in pancreatic adenocarcinoma and RCC [34,56]. In addition, high serum *IFI16* levels are associated with poor overall survival in breast cancer [55]. Using an online database, Wang et al. reported that high *IFI16* expression in ESCC is significantly associated with poor disease-free survival [26]. The results are generally similar to ours, but the high proportion of early-stage cancers in our cohort may account for the lack of statistical significance.

Our study has some limitations. Firstly, we were unable to evaluate the effect of tumor-induced IL-1α on the phenotype and polarization of macrophages in the ESCC microenvironment. IL-1α secreted from lung and gastric cancer cell lines has been reported to promote macrophage infiltration [65,66]. IL-1α derived from ESCC cells may have similar effects on macrophages. However, there are only a few reports on the effect of tumor-secreted IL-1α on macrophages. Secondly, we did not perform in vivo experiments on mice to validate our findings in the present study. A previous report demonstrated that the transplantation of a pancreatic cancer cell line overexpressing *IFI16* into mice promoted TAM infiltration, which in turn promoted tumor growth [34]. In addition, transplantation of the *IFI16*-knockdown ESCC cell line into mice suppresses tumor growth [26]. Finally, the number of resected ESCC samples used in this study was relatively small, and a larger sample size would enable further evaluation of the association between *IFI16* and prognosis.

## 5. Conclusions

In this study, we observed that direct interaction with macrophages elevates *IFI16* expression in ESCC cells. This increase in *IFI16* expression fostered malignant characteristics, including the survival, growth, and migration of ESCC cells, which were mediated by NF-κB and Erk signaling pathways and partially attributable to the regulation of IL-1α expression. Our study is the first to uncover the connection between *IFI16* and IL-1α in the progression of ESCC, suggesting that *IFI16* could serve as a potential prognostic marker for ESCC.

## Figures and Tables

**Figure 1 cells-12-02603-f001:**
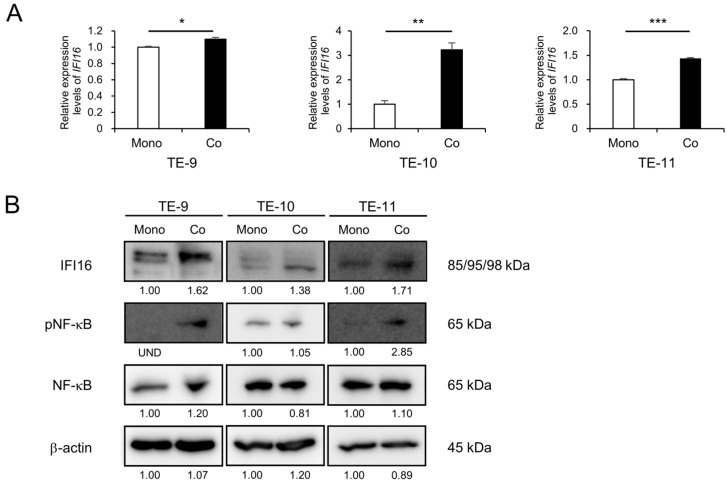
Direct co-culture with macrophages upregulates the expression level of *IFI16* and promotes NF-κB signaling in ESCC cell lines. (**A**) qRT-PCR results showing that upregulated mRNA levels of *IFI16* were observed in co-cultured ESCC cells compared to mono-cultured ESCC cells. *GAPDH* was quantified as an internal control. (**B**) Upregulated protein levels of *IFI16* and promoted phosphorylation of NF-κB in co-cultured ESCC cells compared to mono-cultured ESCC cells, shown using Western blotting. The internal control for Western blotting was β-actin. The expression levels were quantified using ImageJ software, and the relative value was set as 1.00 for mono-cultured ESCC cells. Mono, mono-cultured; Co, co-cultured; UND, undetected. Data are presented as the mean ± SEM of triplicate experiments. * *p* < 0.05, ** *p* < 0.01, *** *p* < 0.001.

**Figure 2 cells-12-02603-f002:**
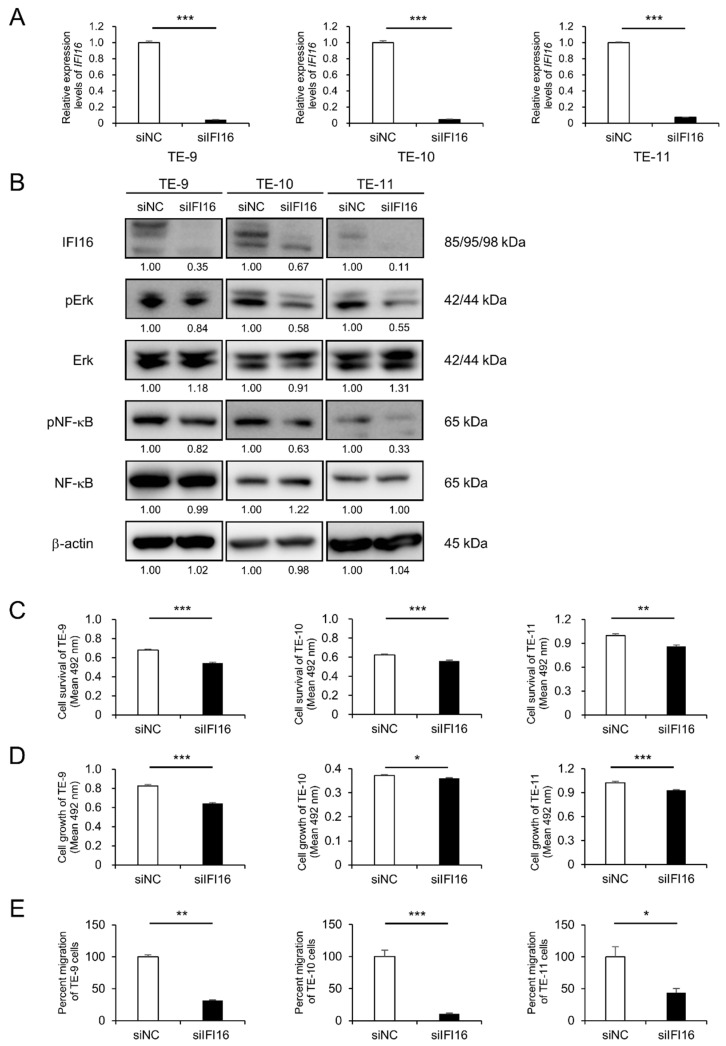
Knockdown of *IFI16* suppresses malignant phenotypes via Erk and NF-κB signaling in ESCC cell lines. (**A**) qRT-PCR was applied to confirm the knockdown of *IFI16* mRNA in ESCC cells. *GAPDH* was quantified as an internal control. (**B**) Western blotting was applied to confirm the knockdown of *IFI16* at the protein level and to evaluate the effect of *IFI16* silencing on the phosphorylation levels of Erk and NF-κB in ESCC cells. The internal control for Western blotting was β-actin. The expression levels were quantified using ImageJ software, and the relative value was set as 1.00 for siNC-transfected ESCC cells. (**C**–**E**) An MTS assay or transwell migration assay revealed suppressed survival (**C**), growth (**D**), and migration (**E**) following *IFI16* knockdown in ESCC cells. Data are presented as the mean ± SEM of triplicate experiments. * *p* < 0.05, ** *p* < 0.01, *** *p* < 0.001. siNC, negative control of siRNA; si*IFI16*, siRNA against *IFI16*.

**Figure 3 cells-12-02603-f003:**
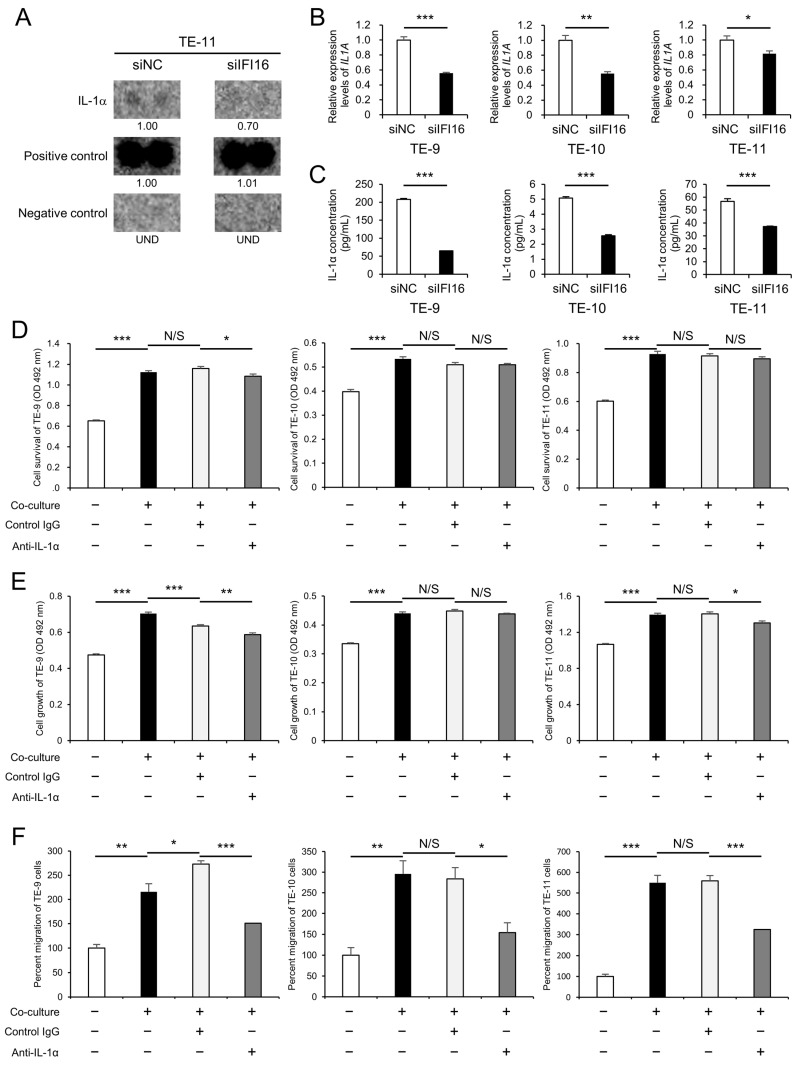
*IFI16* regulates the secretion of IL-1α from ESCC cell lines, which plays an important role in their malignant phenotypes. (**A**) The cytokine array between the culture supernatant of TE-11 cells transfected with siNC and TE-11 cells transfected with si*IFI16* revealed suppressed expression of IL-1α by silencing *IFI16*. The positive and negative control spots were also shown. The expression levels were quantified using ImageJ software, and the relative value was set as 1.00 for siNC-transfected TE-11 cells. (**B**) qRT-PCR was applied to confirm the suppression of *IL1A* mRNA by *IFI16* knockdown in ESCC cells. *GAPDH* was quantified as an internal control. (**C**) ELISA was applied to confirm the suppressed secretion of IL-1α by *IFI16* knockdown in ESCC cells. (**D**–**F**) An MTS assay or transwell migration assay was performed between mono-cultured ESCC cells, co-cultured ESCC cells, co-cultured ESCC cells with a control goat IgG antibody, and co-cultured ESCC cells with an anti-IL-1α neutralizing antibody to evaluate survival (**D**), growth (**E**), and migration (**F**). These phenotypes of ESCC cells after co-culture with macrophages were abrogated by the use of anti-IL-1α neutralizing antibodies. Data are presented as the mean ± SEM of triplicate experiments. N/S, not significant; * *p* < 0.05, ** *p* < 0.01, *** *p* < 0.001. siNC, negative control of siRNA; si*IFI16*, siRNA against *IFI16*; UND, undetected; Control IgG, normal goat IgG control; anti-IL-1α, anti-IL-1α neutralizing antibody.

**Figure 4 cells-12-02603-f004:**
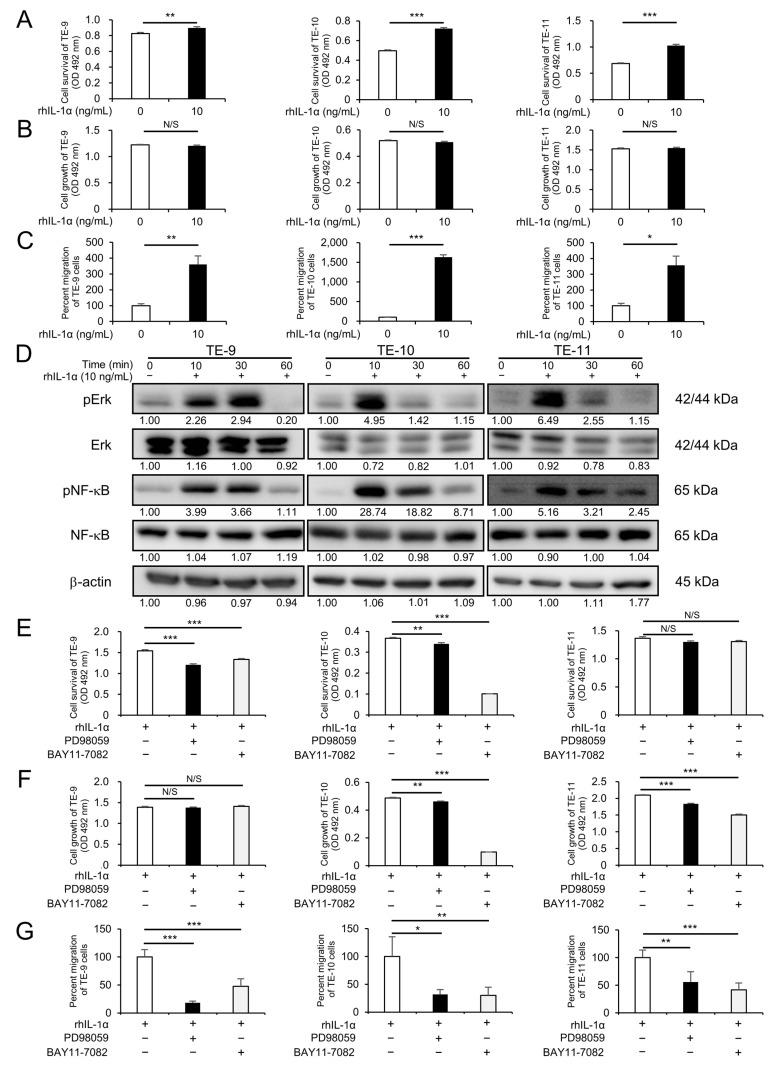
IL-1α enhances malignant phenotypes of ESCC cells via Erk and NF-κB signaling. (**A**–**C**) MTS assay or transwell migration assay revealed enhanced survival (**A**), growth (**B**), and migration (**C**) in ESCC cells after treatment with recombinant human IL-1α (rhIL-1α). (**D**) Promoted phosphorylation levels of Erk and NF-κB in ESCC cells with rhIL-1α were shown by Western blotting. The internal control for Western blotting was β-actin. The expression levels were quantified using ImageJ software, and the relative value was set as 1.00 for rhIL-1α-untreated cells. (**E**–**G**) An MTS assay or transwell migration assay was performed to evaluate the effect against survival (**E**), growth (**F**), and migration (**G**) by inhibiting Erk or NF-κB signaling in ESCC cells treated with an rhIL-1α and Erk inhibitor (PD98059) or NF-κB inhibitor (BAY11-7082). The enhanced malignant phenotypes of ESCC cells by rhIL-1α were abrogated by the use of PD98059 or BAY11-7082. Data are presented as the mean ± SEM of triplicate experiments. N/S, not significant; * *p* < 0.05, ** *p* < 0.01, *** *p* < 0.001.

**Figure 5 cells-12-02603-f005:**
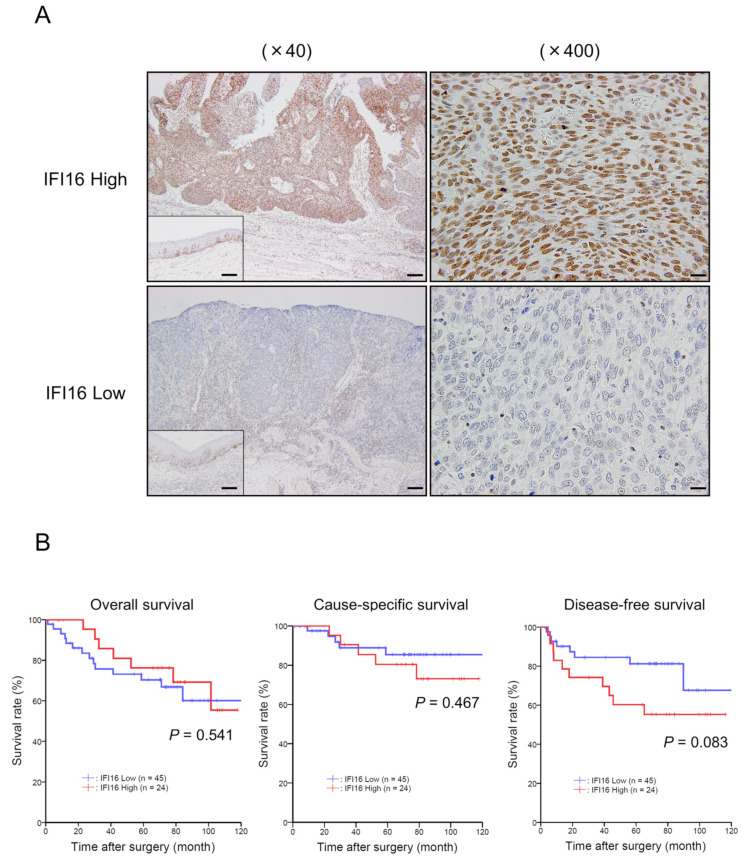
Patients with ESCC who exhibit a high expression of *IFI16* tend to have a poor prognosis in terms of disease-free survival. (**A**) Immunohistochemical staining for *IFI16* was performed in surgically resected ESCC tissues. Representative images for the invasive front of ESCC tissues are shown. Scale bar in ×40 images: 200 μm; Scale bar in ×400 images: 20 μm. (**B**) The survival curve for overall survival, cause-specific survival, and disease-free survival was plotted with the Kaplan–Meier method. The data were analyzed with the log-rank test.

**Figure 6 cells-12-02603-f006:**
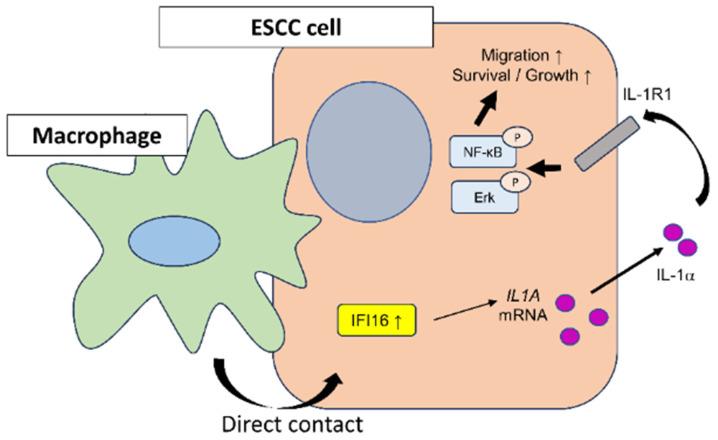
Schematic summary of our findings on the role of *IFI16* in the ESCC microenvironment. *IFI16* was upregulated in ESCC cells which directly interacted with macrophages. Upregulation of *IFI16* led to promoted secretion of IL-1α. IL-1α enhanced malignant phenotypes of ESCC cells, especially migration via Erk and NF-κB signaling.

**Table 1 cells-12-02603-t001:** Gene expressions of HIN-200 family from the cDNA microarray between mono-culture TE-11 cells and TE-11 cells co-cultured with macrophages.

Probe ID	Accession Number	Gene Description		Global Normalization		Ratio
			Symbol	TE-11 Mono	TE-11 Co	(TE11 Co/TE-11 Mono)
H300020876	XM_006711290.1	interferon, gamma-inducible protein 16	*IFI16*	391	1394	3.57
AHsV10000067	XM_006711290.1	interferon, gamma-inducible protein 16	*IFI16*	992	3298	3.32
H200013910	NM_002432.1	myeloid cell nuclear differentiation antigen	*MNDA*	1.38	2.69	1.94
AHsV10000195	NM_004833.1	absent in melanoma 2	*AIM2*	-	-	-
H200011351	NM_004833.1	absent in melanoma 2	*AIM2*	-	-	-
opHsV0400005860	XM_005244930.1	pyrin and HIN domain family member 1	*PYHIN1*	-	-	-

**Table 2 cells-12-02603-t002:** Correlation between clinicopathological data of patients with ESCC and expression of *IFI16* in the invasive front of the tumor.

Variable	Cases	Expression Level of *IFI16* ^a^		*p* Value
		Low (n = 45)	High (n = 24)	
Age, years				
<65	32	17	15	0.050
≥65	37	28	9	
Sex				
Male	55	36	19	1.000
Female	14	9	5	
Histological grade ^b^				
HGIEN + WDSCC	15	11	4	0.456
MDSCC + PDSCC	54	34	20	
Depth of tumor invasion ^b^				
T1	48	35	13	0.042 *
T2, 3	21	10	11	
Lymphatic vessel invasion ^b^				
Negative	37	27	10	0.146
Positive	32	18	14	
Blood vessel invasion ^b^				
Negative	43	28	15	0.982
Positive	26	17	9	
Lymph node metastasis ^b^				
Negative	43	32	11	0.039 *
Positive	26	13	13	
Stage ^c^				
0, I	38	28	10	0.102
II, III, IV	31	17	14	
Expression level of CD68 ^d^				
Low	35	28	7	0.009 **
High	34	17	17	
Expression level of CD163 ^d^				
Low	34	26	8	0.053
High	35	19	16	
Expression level of CD204 ^d^				
Low	34	27	7	0.015 *
High	35	18	17	

^a^ The ESCC samples were divided into two groups (high or low) by their immunohistochemical intensities of *IFI16* in the invasive front of the tumor. ^b^ Based on the 10th edition of the Japanese Classification of Esophageal Cancer [30,31]: HGIEN, high-grade intraepithelial neoplasia; WDSCC, well-differentiated squamous cell carcinoma; MDSCC, moderately differentiated squamous cell carcinoma; PDSCC, poorly differentiated squamous cell carcinoma. T1, tumor invades from the superficial layer to the submucosa; T2, tumor invades the muscularis propria; T3, tumor invades the adventitia. ^c^ Based on the 7th edition of TNM classification by UICC [32]. ^d^ The median values of CD68-, CD163-, or CD204-positive macrophages in the tumor nests and stroma area were calculated. The patients were divided into the low or high group using the median value [14]. Data were analyzed using the χ^2^-test; * *p* < 0.05, ** *p* < 0.01.

## Data Availability

The data presented in this study are available on request from the corresponding author.

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
