# Peer review of "IFI16 Induced by Direct Interaction between Esophageal Squamous Cell Carcinomas and Macrophages Promotes Tumor Progression via Secretion of IL-1α"

_cells, 2023, doi:10.3390/cells12222603_

Round 1
Reviewer 1 Report
Comments and Suggestions for Authors
In the present manuscript the Authors have addressed the role of the enhanced expression of IFI16 in ESCC cell lines after their co-culture with macrophages. The paper is generally well written although some clarifications need to be made:
A)I propose to better explain that the present study raises from the re-examination of microarray analysis previously performed by the Authors (GSE174796) and from the observation that IFI16 , among the HIN-200 family genes, is significantly enhanced after co-culture of the TE11 cell line with macrophages, as well as the previously reported S100A8/A9, IL7R and MMP9 (20-22). I suggest to postpone lines 87-90 of the introduction in the first paragraph further adding here also the Table I (therefore not showing it as supplementary data).
B) Lines 241-242: Please rephrase: i.e. We have also previously demonstrated increased ERK1/2 phosphorylation in macrophages/ESSC cell lines co-cultures (data not shown) (21)
C) Please place the figure 2 legend immediately below the related figure.
In addition, how many experiments have been performed for data presented in figura 2 C, D, E …? they are the mean± SEM of N..experiments? Please add these informations.
D) Lines 277-278 and Figure 3: Although data reported in figures 3B and C clearly demonstrate decreased levels of IL1-α-mRNA and of this released cytokine by siIF16I-transfected cells, it seems very difficult to me to see any difference between si-NC- and siIF16I-transfected ESCC cells from the IL1-α spots shown in fig. 3. Looking to your previous paper published in Cancers (cancers,2023 ,15) I may note that here the constitutive release of IL-1α is clearly detectable, at difference with Figure 3 here presented. Possibly you may provide a better figure. Otherwise I would change lines 277-282: perhaps you may say that “although a very faint decrease of IL1-α was detected by cytokine array analysis, quantitative RT-PCR and Elisa assays demonstrated a significant decrease of…….. “ . Please also change lines 403-404 in the Discussion.
E) malignant phenotypes??? Can you please to explain better what do you mean with “malignant phenotype”? I guess functional and phenotypical features linked to chemotaxis and metastatic invasion, but I suggest to better explain it at least when you mention the “malignant phenotype” for the first time.
F) Figure legend 4. Please indicate how many experiments have been performed for figures A, B, C, E,F,G thus indicating that the reported data are the mean±SD of N…. experiments.
G)Discussion: The Discussion needs to be re-organized making it more fluent and readable.
I propose to substitute lines 400-402 with: “In this study we have demonstrated that the expression of IFI16 was enhanced in ESCC cell lines co-cultured with macrophages. Moreover, IFI16 regulated the secretion of IL-1α by ESCC cells…… and so on with your presented results and references….( lines 403-431). Then you may better discuss the paper from Wang and collaborators (ref.26 here reported) that describes IFI16 as a metastatic-related protein. Here the IFI16 knocking down induced downregulation of FGF proteins in 30M cells, used as a model of ESCC metastasis. I would implement therefore the description of the Wang reported results, further discussing better the many molecules potentially involved in metastasis and invasion that could represent therapeutical targets.
-line450: I would substitute “identified” with “ have shown”
Comments on the Quality of English LanguageMinor English editing
Reviewer 2 Report
Comments and Suggestions for Authors
The study entitled “IFI16 Induced by Direct Interaction between Esophageal Squamous Cell Carcinomas and Macrophages Promotes Tumor Progression via Secretion of IL-1α”. Here are my comments:
1. The abstract may benefit from modification due to many abbreviations that may impair comprehension of the manuscript's objectives for a non-specialist audience. The authors must briefly describe the research's contextual underpinning in the abstract section. What variables influenced the researcher's decision to conduct this study? What importance does this work have?
2. It is suggested that authors supplement the literature offered by including a detailed author introduction. The introduction should be revised using the most recent literature sources.
3. What calculating method was employed for the analysis of RT-PCR?
4. The discussion section should begin with a brief paragraph describing the research's objectives and significant discoveries. It is best to provide more data to back up your hypotheses.
5. The histological pictures must be clarified to highlight the difference between the two groups, and a more extensive analysis must be undertaken.
6. Please show individual data points of all figures.
7. In light of the findings, it is strongly advised that the writers revise the conclusion section.
8. The discussion forum should discuss the limits encountered during the study process. The authors are expected to provide a comprehensive discussion of the limits of their study, therefore elucidating the scientific research community.
Round 2
Reviewer 2 Report
Comments and Suggestions for Authors
The authors have answered my questions, and the paper has significantly improved.